# Low Vitamin K Status Is Associated with Increased Elastin Degradation in Chronic Obstructive Pulmonary Disease

**DOI:** 10.3390/jcm8081116

**Published:** 2019-07-27

**Authors:** Ianthe Piscaer, Jody M. W. van den Ouweland, Kristina Vermeersch, Niki L. Reynaert, Frits M. E. Franssen, Spencer Keene, Emiel F. M. Wouters, Wim Janssens, Cees Vermeer, Rob Janssen

**Affiliations:** 1Department of Respiratory Medicine, Maastricht University Medical Center, 6229 HX Maastricht, The Netherlands; 2Clinical Chemistry, Canisius-Wilhelmina Hospital, 6532 SZ Nijmegen, The Netherlands; 3Laboratory of Respiratory Diseases, Catholic University Leuven, 3000 Leuven, Belgium; 4CIRO, Centre of Expertise for Chronic Organ Failure, 6085 NM Horn, The Netherlands; 5Department of Respiratory Medicine, University of Birmingham, Birmingham B15 2TT, UK; 6Department of Respiratory Medicine, Catholic University Leuven, 3000 Leuven, Belgium; 7R&D Group VitaK, Maastricht University, 6229 EV Maastricht, The Netherlands; 8Department of Pulmonary Medicine, Canisius-Wilhelmina Hospital, 6532 SZ Nijmegen, The Netherlands

**Keywords:** chronic obstructive pulmonary disease, desmosine, elastin degradation, matrix Gla protein, vitamin K

## Abstract

Elastin degradation is accelerated in chronic obstructive pulmonary disease (COPD) and is partially regulated by Matrix Gla Protein (MGP), via a vitamin K-dependent pathway. The aim was to assess vitamin K status in COPD as well as associations between vitamin K status, elastin degradation, lung function parameters and mortality. A total of 192 COPD patients and 186 age-matched controls were included. In addition to this, 290 COPD patients from a second independent longitudinal cohort were also included. Vitamin K status was assessed by measuring plasma inactive MGP levels and rates of elastin degradation by measuring plasma desmosine levels. Reduced vitamin K status was found in COPD patients compared to smoking controls (*p* < 0.0005) and controls who had never smoked (*p* = 0.001). Vitamin K status was inversely associated with desmosine (cohort 1: *p* = 0.001; cohort 2: *p* = 0.004). Only few significant associations between vitamin K status and lung function parameters were found. Mortality was higher in COPD patients within the quartile with the lowest vitamin K status compared to those within the other quartiles (hazard ratio 1.85, 95% confidence interval (CI), 1.21–2.83, *p* = 0.005). In conclusion, we demonstrated reduced vitamin K status in COPD and an inverse association between vitamin K status and elastin degradation rate. Our results therefore suggest a potential role of vitamin K in COPD pathogenesis.

## 1. Introduction

Chronic obstructive pulmonary disease (COPD) is defined by a combination of chronic respiratory symptoms and persistent airflow limitation, caused by small airways disease, parenchymal destruction (emphysema) or a combination of both [1]. COPD pathogenesis is characterized by chronic inflammation and protease–antiprotease imbalance [1]. Although smoking is indisputably the most important risk factor for this disease in the Western world [1], it is still unclear why some smokers develop COPD whereas others with comparable smoking history do not. Genetic variance offers only little explanation, and therefore environmental factors beyond smoking —the exposome—should be addressed.

Accumulating evidence suggests that the fat-soluble vitamins A and D are implicated in the pathogenesis of COPD [2,3,4]. Vitamin K also belongs to the fat-soluble vitamins; however, its role in COPD has never been explored. Vitamin K is important for the activation of clotting factors in the liver but is also an essential cofactor in the activation of various other proteins [5]. Matrix Gla Protein (MGP) is a vitamin K-dependent protein and one of the few potent inhibitors of elastin calcification [6], since other anti-calcifying proteins, such as fetuin, are too large to enter the interior of the elastin fibres [6,7]. Arterial calcifications start in elastin fibres [8] and can be induced in rats through the administration of vitamin K antagonists (VKAs) by preventing MGP activation [9]. The mineralization of elastin results in enhanced proteolytic elastin degradation, given that protease synthesis increases parallel to the calcium content of elastin fibres [10]. An animal model of elastocalcinosis has suggested that MGP also has protective properties against elastin degradation [9].

Numerous studies have focused on the role of MGP in the vascular wall [6]. However, MGP is also highly expressed in lungs [11]. Elastin is an important component of both lungs and arteries, and, although the arrangement of elastic fibres is different for each type of tissue [12], the elastin protein itself has comparable chemical properties. It is therefore expected that the previously demonstrated favourable properties of MGP [6] are not only applicable to blood vessels, but also to lung tissue. The rate of elastin degradation is accelerated in COPD and related to mortality [13]. Therefore, we hypothesized that vitamin K plays a role in COPD pathogenesis through MGP modulation (Figure 1).

The aim of this study was to compare vitamin K status between subjects with COPD and healthy controls as well as to evaluate the association of vitamin K status with the rate of elastin degradation, lung function parameters and mortality.

## 2. Experimental Section

### 2.1. Study Population and Design

A total of 192 COPD patients and 186 age-matched controls from the Individualized COPD Evaluation in relation to Ageing (ICE-Age) study were included (www.controlled-trials.com, identifier ISRCTN86049077). Clinically stable COPD patients aged 50 to 75 years who had been admitted for pulmonary rehabilitation in CIRO, Centre of Expertise for Chronic Organ Failure, (Horn, The Netherlands) were enrolled between 1 October, 2010, and 1 October, 2014. Healthy controls were recruited from the same region. Former smoking controls were regarded as smokers in order not to underestimate the effect of smoking.

We also included patients from a second prospective cohort (the Leuven COPD study) to assess the effects of vitamin K status on mortality and in order to provide sufficient patient heterogeneity for assessing the effects of vitamin K status on lung function parameters and elastin degradation. From this cohort, 290 COPD patients, aged 50 years or older with a history of ≥15 smoking pack years were selected, matched for age and gender within each GOLD category. Subjects were recruited from the University Hospitals Leuven (Belgium) between 1 October, 2007, and 2 September, 2013. Exclusion criteria were recent diagnosis of a malignancy, other respiratory diseases than COPD, history of significant thoracic surgery or organ transplantation. Details of both studies have previously been described [14,15].

Both studies were conducted according to the guidelines laid down in the Declaration of Helsinki and good clinical practice guidelines. Written informed consent was obtained from all subjects before entering the study. The ICE-Age and Leuven COPD study were approved by the local ethics review board of the Maastricht University Medical Centre (10-3-033) and the University Hospitals Leuven (S50623), respectively.

Both cohorts were analysed independently, given that the ICE-Age study control group was not age-matched with COPD subjects in the Leuven cohort, and the Leuven cohort provided follow-up data, while the ICE-Age cohort was cross-sectional in nature. Patient characteristics of both cohorts are presented in Table 1 and Table 2.

### 2.2. Assessments and Follow-up

Blood samples were taken in both cohorts. Spirometry was performed and static lung volumes and diffusion capacity were measured according to the American Thoracic Society/European Respiratory Society guidelines in all subjects from both cohorts. Details about data collection have previously been described [14,15]. In the Leuven study, the presence of aortic calcification was scored at the level of the ascending aorta, aortic arch and descending aorta. Aortic calcification score could range from 0 (no calcifications) to 3 (calcifications at all levels). Follow-up data regarding mortality up until 7 years were available for subjects in the Leuven cohort.

### 2.3. Quantification of Vitamin K Status

Measuring inactive levels of vitamin K-dependent proteins in the circulation is assumed to be the most appropriate method to assess total vitamin K status [16]. In both cohorts, desphospho-uncarboxylated (dp-uc; i.e., inactive) MGP levels were used as surrogate marker for vitamin K status, which are inversely correlated with extrahepatic vitamin K status. Plasma dp-ucMGP was measured using a dual-antibody test based on the previously described sandwich ELISA developed by VitaK (Maastricht, The Netherlands) [17]. The intra- and inter-assay variations of this ELISA are 7.6% and 6.8%, respectively, and its sensitivity is 50 pmol/L. The within-subject variability of dp-ucMGP is very limited [17].

### 2.4. Quantification of Elastin Degradation

The rate of elastin degradation was quantified by measuring plasma (p) desmosine and isodesmosine (together referred to as DES) levels. These amino acids only occur in crosslinked elastin, and their plasma levels are therefore a reflection for the rate of elastin degradation. Isodesmosine and desmosine fractions were measured separately by liquid chromatography-tandem mass spectrometry, as previously described, using deuterium-labelled desmosine as internal standard [18,19]. Coefficient of variations of intra- and inter-assay imprecision were <10%, lower limit of quantification 140 ng/L and assay linearity up to 52,000 ng/L.

### 2.5. Statistical Analysis

SPSS (version 24, IBM, Chicago, IL, USA) was used for statistical analyses. We used *p* < 0.05 as the threshold for statistical significance. Data are presented as estimated marginal means and 95% confidence intervals (CI)s. Lung function parameters were used as a percentage of the predicted value for the analyses [20].

Analysis of covariance (ANCOVA) was used to assess both differences in mean dp-ucMGP and pDES between COPD patients and control groups, adjusted for age, gender, body mass index (BMI), smoking packyears and VKA use. Given the large effects of VKAs on vitamin K status [21], differences in dp-ucMGP levels between COPD and control subjects were also assessed in subjects not using VKAs.

Multivariable linear regression analysis was performed to assess the association between VKA use/baseline characteristics (i.e., age, gender, BMI and smoking packyears) and both dp-ucMGP and pDES (dependent variables); the presence/absence of COPD and FEV1 were included as covariates. The association between dp-ucMGP and pDES was subsequently corrected for all these variables.

Associations between dp-ucMGP, lung function parameters and GOLD stage were assessed in the subgroup of COPD patients using multivariable linear regression corrected for age, gender, BMI, smoking packyears and VKA use.

Data with a log-normal (LN) distribution were log-transformed prior to the ANCOVA and multivariable linear regression analyses. Means and 95% CIs of log-transformed variables were back-transformed into geometric means and 95% CIs. The expected ratios of the logged variables were presented for associations between two continuous variables in which the dependent variable was log-transformed, i.e., the expected change in the dependent variable when the independent variable increases by 10% (in case of a log-transformed independent variable) or increases by 10 percent points or 1 unit (in case of a non-log-transformed independent variable).

To assess the association between vitamin K status and mortality, subjects from the Leuven cohort in which follow-up data were available were divided into four equally sized groups based on dp-ucMGP levels (i.e., quartiles). Cox regression analysis was used to compare survival between subjects in those vitamin K quartiles, and to assess the association of pDES with survival. Mortality was corrected for age, gender, BMI, number of smoking pack-years, VKA use, aortic calcification, MRC, and for those lung function tests that had a mutual correlation coefficient with forced expiratory volume in 1 s (FEV1) below 0.65, in order to avoid overcorrection for strongly correlated covariates.

## 3. Results

### 3.1. ICE-Age Cohort

#### 3.1.1. Vitamin K Status in the ICE-Age Cohort

In the ICE-Age cohort, dp-ucMGP levels had a LN distribution and were therefore log-transformed prior to analysis. Dp-ucMGP was significantly higher in COPD patients (1127 pmol/L, 95% CI, 996 to 1274 pmol/L) compared to (former) smoking controls (821 pmol/L, 95% CI, 716 to 939 pmol/L, *p* < 0.0005) and controls who had never smoked (841 pmol/L, 95% CI, 709 to 997 pmol/L, *p* = 0.001, Figure 2). No statistical difference was found between (former) smoking controls and controls who had never smoked (*p* = 0.742). After exclusion of VKA users (*n* = 17) and subjects in which VKA use was unknown (*n* = 6), dp-ucMGP remained significantly higher in COPD patients (650 pmol/L, 95% CI, 601 to 703 pmol/L) compared to smoking controls (472 pmol/L, 95% CI, 433 to 513 pmol/L, *p* < 0.0005) and controls who had never smoked (484 pmol/L, 95% CI, 424 to 552 pmol/L, *p* = 0.001); and no statistical difference was found between smoking controls and controls who had never smoked (*p* = 0.726).

Dp-ucMGP was significantly higher in users of VKAs (1664 pmol/L, 95% CI, 1337 to 2071 pmol/L) compared to non-VKA users (557 pmol/L, 95% CI, 532 to 584 pmol/L, *p* < 0.0005). Dp-ucMGP was positively associated with age (2.3%, 95% CI, 1.6 to 3.0%, per year increase in age, *p* < 0.0005) and BMI (3.6%, 95% CI, 2.5 to 4.7%, per kg/m^2^ increase in BMI, *p* < 0.0005), but was not associated with gender (*p* = 0.064) or smoking pack-years (*p* = 0.489).

In COPD patients, dp-ucMGP was not associated with GOLD stage (*p* = 0.357). Dp-ucMGP was inversely associated with diffusion capacity of the lung for carbon monoxide (DLCO) (−4.9%, 95% CI, −8.6 to −2.0%, per 10 percent points increase in DLCO, *p* = 0.005), but not with other lung function parameters (FEV1, forced vital capacity (FVC), FEV1/FVC, inspiratory FVC, Krogh factor, residual volume (RV) and intrathoracic gas volume (ITGV)).

#### 3.1.2. Elastin Degradation in the ICE-Age Cohort

pDES levels had a LN distribution and were therefore log-transformed prior to analysis. Dp-ucMGP was significantly and positively associated with pDES (0.96%, 95% CI, 0.41 to 1.50%, per 10% increase in dp-ucMGP, *p* = 0.001, Figure 3a), i.e., lower vitamin K status was associated with accelerated elastin degradation. pDES was significantly higher in COPD patients (341 ng/L, 95% CI, 319 to 365 ng/L) compared to (former) smoking controls (294 ng/L, 95% CI, 272 to 317 ng/L, *p* < 0.0005) and controls who had never smoked (300 ng/L, 95% CI, 273 to 330 ng/L, *p* = 0.007). No significant differences in pDES levels were found between (former) smoking controls and controls who had never smoked (*p* = 0.584).

pDES was not significantly associated with use of VKAs (*p* = 0.522), gender (*p* = 0.064), BMI (*p* = 0.794) or smoking packyears (*p* = 0.132). A positive association was found between age and pDES (2.1%, 95% CI, 1.7 to 2.5%, per year increase in age, *p* < 0.0005).

In subjects with COPD, GOLD stage was not associated with pDES (*p* = 0.147). pDES was significantly and inversely associated with DLCO (−3.9%, 95% CI −5.8 to −1.0%, per 10 percent points increase in DLCO, *p* = 0.004) and Krogh factor (−3.0, 95% CI, −4.9 to −1.0%, per 10 percent points increase in Krogh factor, *p* = 0.004). pDES was not associated with other lung function parameters (FEV1, FEV1/FVC, FVC, inspiratory FVC, RV and ITGV).

### 3.2. Leuven COPD Cohort

#### 3.2.1. Vitamin K Status in the Leuven COPD Cohort

Dp-ucMGP levels had a LN distribution and were therefore log-transformed prior to analysis. Dp-ucMGP was not significantly associated with GOLD stage (*p* = 0.793). Dp-ucMGP levels were significantly higher in COPD subjects using VKAs (1260 pmol/L, 95% CI, 858 to 1848 pmol/L) than in subjects not using VKAs 523 pmol/L, 95% CI, 455 to 599 pmol/L, *p* < 0.0005). Dp-ucMGP was not significantly associated with age (*p* = 0.079), gender (*p* = 0.336), BMI (*p* = 0.109) or smoking pack-years (*p* = 0.845). Dp-ucMGP was inversely associated with RV (−3.0%, 95% CI, −4.9 to −1.0%, per 10 percent point increase in RV, *p* = 0.012, Table 3), ITGV (−3.9, 95% CI, −6.8 to −1.0%, per 10 percent points increase in ITGV, *p* = 0.023) and total lung capacity (TLC) (−9.5%, 95% CI, −15.6 to −3.0%, per 10 percent points increase in TLC, *p* = 0.004). No significant association was found between dp-ucMGP and other lung function parameters (i.e., specific airway conductance, FEV1, FEV1/FVC, airway resistance, FVC, vital capacity (VC), DLCO and Krogh factor).

#### 3.2.2. Elastin Degradation in the Leuven COPD Cohort

pDES levels had a LN distribution and were therefore log-transformed prior to analysis. In the Leuven cohort, there was a significant and positive association between dp-ucMGP and pDES (0.62%, 95% CI, 0.20 to 1.1%, increase in pDES when dp-ucMGP increases by 10%, *p* = 0.004), i.e., an inverse association between vitamin K status and elastin degradation was found (Figure 3b). pDES was significantly lower in GOLD stage IV (327 ng/L, 95% CI, 293-366 ng/L) compared to GOLD II (417 ng/L, 95% CI, 376 to 463 ng/L, *p* < 0.0005), but not compared to GOLD I (370 ng/L, 95% CI, 330 to 414 ng/L, *p* = 0.051) or GOLD III (368 ng/L, 95% CI, 330 to 411 ng/L, *p* = 0.057). No significant association was found between pDES and use of VKAs (*p* = 0.485), BMI (*p* = 0.294), smoking pack-years (*p* = 0.065) or gender (*p* = 0.360). pDES was positively associated with age (3.0%, 95% CI, 2.5 to 3.7%, per year increase in age, *p* < 0.0005).

pDES was significantly and positively associated with FEV1 (2.0%, 95% CI, 0.0 to 4.1%, per 10 percent points increase in FEV1, *p* = 0.027), specific airway conductance (2.0%, 95% CI, 0.0 to 3.0%, per 10 percent points increase in specific airway conductance, *p* = 0.009), and significantly and inversely associated with Krogh factor (−2.0%, 95% CI, −3.9 to −0.02%, per 10 percent points increase in Krogh factor, *p* = 0.048), RV (−1.0%, 95% CI −2.0 to −1.0%, per 10 percent point increase in RV, *p* = 0.001), ITGV (−2.0%, 95% CI −3.0 to −1.0%, per 10 percent points increase in ITGV, *p* = 0.002), FEV1/FVC (5.1%, 95% CI, 2.0 to 9.4%, per 10 percent points increase in FEV1/FVC, *p* = 0.001) and TLC (−4.9%, 95% CI −6.8 to −2.0%, per 10 percent points increase in TLC, *p* = 0.001). pDES was not associated with FVC, DLCO, airway resistance and VC.

#### 3.2.3. Effects of Vitamin K Status on Mortality in the Leuven COPD Cohort

Follow-up data were available in 283 subjects, with a mean follow-up time of 5.3 years (standard deviation 2.5 years). Out of those, 134 (47.3%) died during follow-up. Lung function parameters Krogh factor, ITGV and TLC had a mutual correlation coefficient with FEV1 below 0.65 and were therefore used as covariates in addition to the aforementioned covariates. Subjects were divided into four quartiles (Q) based on dp-ucMGP: Q1 dp-ucMGP 15–405 pmol/L, Q2 dp-ucMGP 405–641 pmol/L, Q3 dp-ucMGP 641–909 pmol/L and Q4 dp-ucMGP 909–4965 pmol/L. Compared to Q1, mortality was significantly higher in Q4 (hazard ratio (HR) 1.76, 95% CI, 1.04 to 3.01, *p* = 0.037), while it was not different in Q2 (*p* = 0.783) and Q3 (*p* = 0.828) (Figure 4). Subjects in Q4 had higher mortality rates compared to subjects in the other three quartiles combined (HR 1.85, 95% CI, 1.21 to 2.83, *p* = 0.005). Mortality was significantly higher in subjects using VKAs compared to subjects not using VKAs (HR 1.98, 95% CI, 1.12 to 3.52, *p* = 0.019). In addition to this, pDES was positively associated with mortality (HR 1.001, 95% CI, 1.000 to 1.002, per ng/L increase in pDES, *p* = 0.006).

## 4. Discussion

This study has demonstrated that vitamin K status was reduced in COPD patients compared to age-matched controls. Furthermore, low vitamin K status was associated with accelerated elastin degradation in two independent COPD cohorts and related to decreased survival. Therefore, to our knowledge, this is the first study to suggest that vitamin K might be implicated in COPD pathogenesis.

Elastin degradation was quantified by pDES, which is regarded by the COPD Biomarker Qualification Consortium as a biomarker with great potential [22]. We demonstrated elevated pDES levels in COPD patients, which is consistent with previous studies [13]. It has been postulated that elastin degradation in COPD is not only accelerated in lungs, but that this process is systemically enhanced. This hypothesis is supported by Maclay et al. who demonstrated a correlation between emphysema severity, arterial stiffness and skin wrinkling, all of which result from accelerated elastin degradation [23]. We demonstrated that higher pDES levels were associated with increased mortality in subjects with COPD. This has previously been demonstrated in a large cohort of COPD subjects [13]. Slowing down the rate of elastin degradation might therefore be a novel therapeutic target in COPD patients. Alpha-1 antitrypsin augmentation therapy has already been proven effective in reducing pDES in the subgroup of COPD patients with alpha-1 antitrypsin deficiency [24]. Based on our current results, we postulate that vitamin K supplementation may also decelerate elastin degradation in patients with COPD. A vitamin K intervention trial with change in pDES as primary endpoint is warranted to test this hypothesis.

There are several possible explanations for reduced vitamin K status in COPD. One of them might be low vitamin K consumption. Vitamin K can only be obtained exogenously as either vitamin K1 or vitamin K2 [16]. Cheese is one of the major sources of vitamin K2 in many countries [25] and cheese consumption has been shown to be associated with better lung function and less emphysema in a large observational study [26]. Another potential mechanism could be an interindividual variation in vitamin K recycling. Certain polymorphisms in the gene coding for a protein involved in the vitamin K cycle are associated with reduced vitamin K recycling rates, and those might be overrepresented in COPD patients [27]. Increased vitamin K expenditure might be another explanation [27]. Elastin degradation is accelerated in COPD patients and this stimulates elastin calcification. MGP synthesis will increase in order to counteract this process. This will consequentially raise the demand for vitamin K activation and might induce vitamin K deficiency.

Low vitamin K status was associated with increased mortality in COPD patients, independent from known risk factors such as age and COPD severity. This association has also been found in both patients with vascular and renal disease [28,29]. Interestingly, subjects in the quartile with lowest vitamin K status had higher mortality compared to subjects in the three other quartiles, while mortality among subjects in the three lower quartiles was comparable. This is suggestive for a certain threshold of vitamin K status in COPD, below which the risk for mortality increases. In the present study, COPD subjects who used VKAs had reduced vitamin K status and mortality was significantly increased in this subgroup. However, it should be noted that it was not known whether the use of VKAs was intermittent or chronic. We recently found higher mortality rates in a retrospective study in COPD subjects who used VKAs compared to COPD subjects who used direct oral anticoagulants or no oral anticoagulants [30]. Unfavorable effects of VKAs on survival have previously also been found in patients with idiopathic pulmonary fibrosis [31,32,33,34]. Since vitamin K status was inversely associated with elastin degradation, which is an established predictor of mortality in COPD [13], we propose that the acceleration of elastinolysis might be one possible mechanism responsible for higher mortality in subjects with poor vitamin K status. However, the death causes were unknown in the majority of our study subjects, and we therefore acknowledge that these results should be interpreted with caution. Animal models are required to unravel the true underlying pathogenic mechanisms.

Although low vitamin K status was associated with the presence of COPD, only a few associations were found between vitamin K status and the severity of lung function impairments in COPD patients from both cohorts. However, we would expect that vitamin K status relates to disease activity rather than severity, given that vitamin K status was associated with accelerated loss of elastin, which is an important driver of COPD development and progression [35]. As we did not have follow-up lung function data, we were unable to assess whether vitamin K status correlates with lung function decline and additional studies are therefore needed.

Our study has several strengths, such as the well-characterized study populations, the comprehensiveness of lung function testing, the long follow-up time in the Leuven cohort and the objective quantification of vitamin K status in contrast to the use of subjective food questionnaires. However, there are some limitations that should be noted. Although a causal role for vitamin K deficiency in COPD is conceivable based on the available literature, this conclusion cannot be definitively drawn. In addition, our study is not able to disentangle whether elastin degradation is a marker of vascular disease rather than of lung disease. Longitudinal follow-up studies, designed to evaluate the decline in lung function and CT-scored emphysema, are needed to clarify the relation between loss of lung tissue and vitamin K status. Animal models of cigarette smoke-induced emphysema are needed to assess whether vitamin K might be protective against the development of COPD-like characteristics in the lungs [36]. These studies are essential before clinical trials are undertaken to evaluate the effects of vitamin K supplementation on disease progression and mortality in patients with COPD.

## 5. Conclusions

This is the first study demonstrating reduced vitamin K status in COPD patients as well as an inverse association between vitamin K status and the rate of elastin degradation. Our results therefore suggest that vitamin K may play a role in the pathogenesis of COPD and could be a potential target for intervention.

## Figures and Tables

**Figure 1 jcm-08-01116-f001:**
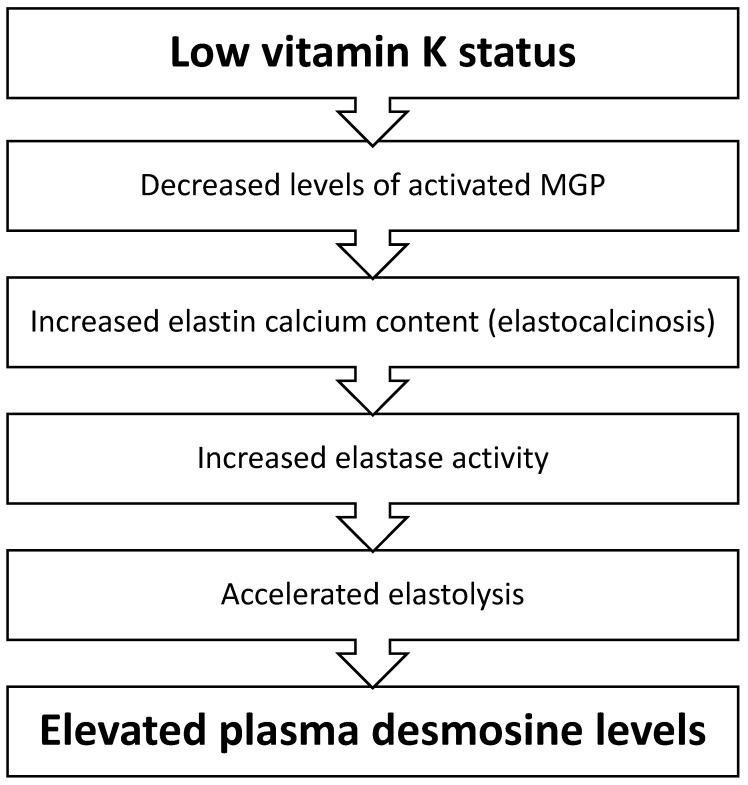
Proposed mechanism by which low vitamin K status leads to the acceleration of elastin degradation reflected by elevated plasma desmosine levels. Low vitamin K status will lead to impaired Matrix Gla Protein (MGP) activation and, subsequently, to an increased calcium content within elastin fibres. Elastin calcification causes elastin degradation. During the process of elastin degradation, desmosine is released from crosslinked elastin fibres and leaks from the extracellular matrix to the blood stream.

**Figure 2 jcm-08-01116-f002:**
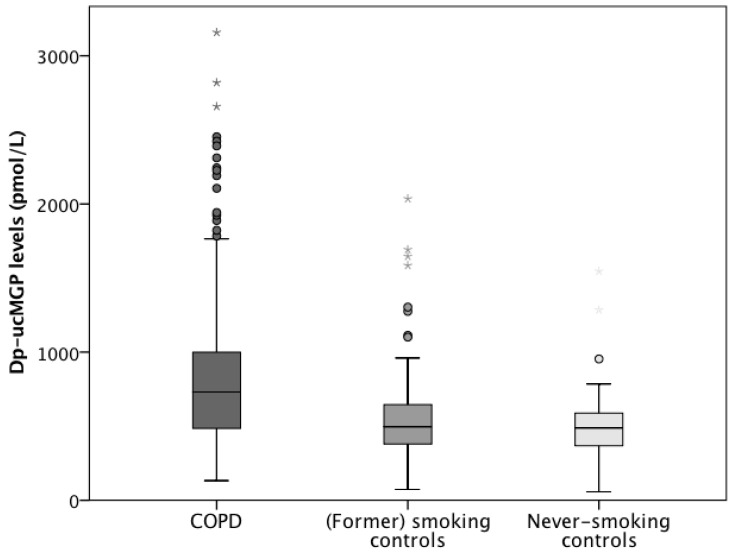
Boxplots (5th percentile, 1st quartile, median, 3rd quartile, and 95th percentile) showing a significant difference in plasma desphospho-uncarboxylated Matrix Gla Protein levels between COPD patients (dark grey), (former) smoking control subjects (middle grey) and never-smoking control subjects (light grey) in the ICE-Age study. Outliers are presented as points and extreme outliers as asterisks.

**Figure 3 jcm-08-01116-f003:**
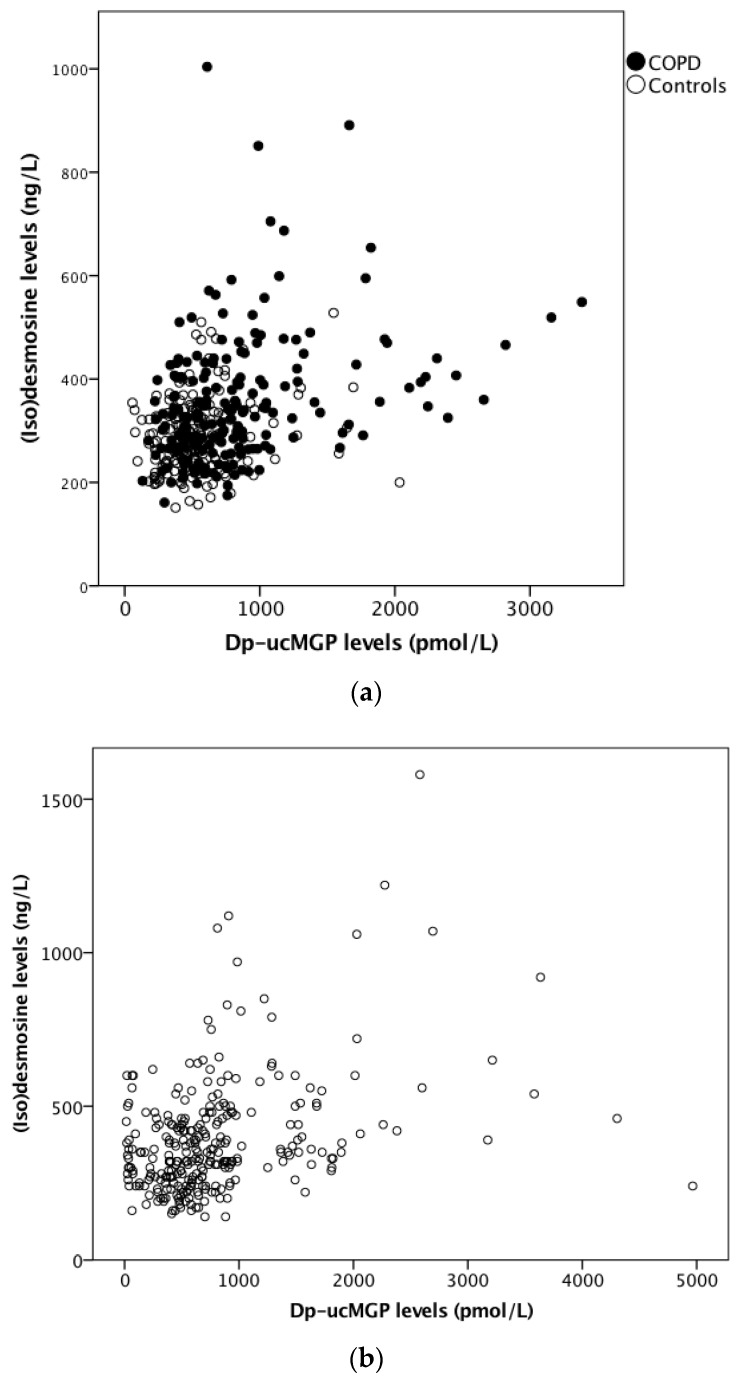
Scatterplots showing (**a**) the significant positive association between plasma inactive Matrix Gla Protein (MGP) levels and plasma desmosine levels in COPD patients (black circles) and controls (white circles) from the ICE-Age cohort, and (**b**) the association between plasma inactive MGP levels and plasma desmosine levels in COPD patients in the Leuven cohort.

**Figure 4 jcm-08-01116-f004:**
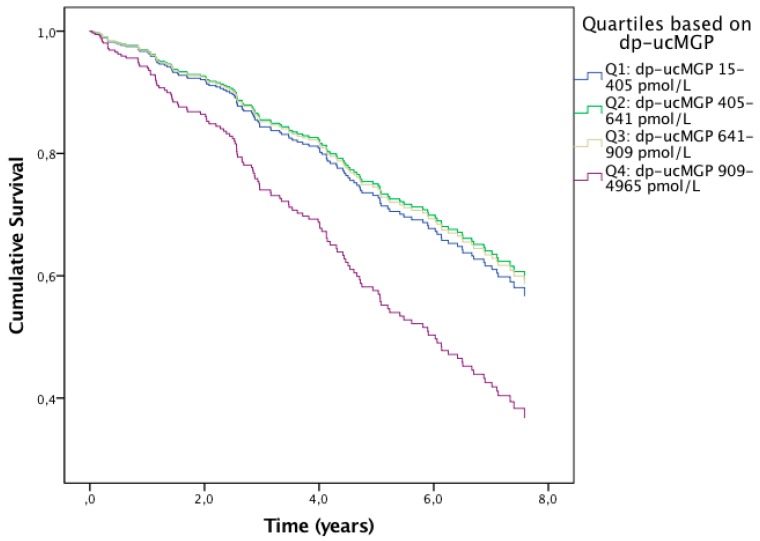
Regression lines showing the cumulative survival over time of subjects from the vitamin K quartiles corrected for age and disease severity. The blue line represents the first quartile (i.e., highest vitamin K status), the green line the second quartile, the beige line the third quartile and the purple line the fourth quartile (i.e., lowest vitamin K status). Mortality was significantly increased in subjects from the fourth quartile compared to subjects from the other three quartiles.

**Table 1 jcm-08-01116-t001:** Patient characteristics of the Individualized COPD Evaluation in relation to Ageing (ICE-Age) cohort. Continuous data are presented as mean ± standard deviation.

ICE-Age Cohort	COPD	(Former) Smoking Controls	Never-Smoking Controls
Subjects, *n*	192	124	62
Age, years	61.9 ± 7.1	61.3 ± 5.9	60.1 ± 7.7
Male, *n (%)*	113 (58.9)	63 (50.8)	23 (37.1)
Smoking status			
Never-smokers, *n (%)*	3 (1.6)		62 (100)
Former smokers, *n (%)*	124 (64.6)	105 (84.7)	
Current smokers, *n (%)*	65 (33.9)	19 (15.3)	
Smoking pack years	49.2 ± 29.7	18.1 ± 14.9	0.0 ± 0.2
BMI, kg/m^2^	26.7 ± 5.3	27.3 ± 3.4	26.0 ± 3.0
VKAs			
No VKAs, *n (%)*	174 (90.6)	120 (96.8)	61 (98.4)
VKA use, *n (%)*	14 (7.3)	3 (2.4)	0 (0)
VKA use unknown, *n (%)*	4 (2.1)	1 (0.8)	1 (1.6)
COPD stage			
COPD GOLD I, *n (%)*	1 (0.5)		
COPD GOLD II, *n (%)*	95 (49.5)		
COPD GOLD III, *n (%)*	77 (40.1)		
COPD GOLD IV, *n (%)*	19 (9.9)		
MRC, points	3.0 ± 1.0		
Lung function values			
FEV1, L	1.39 ± 0.52	3.36 ± 0.70	3.31 ± 0.79
Percent of predictive normal value	49.3 ± 15.7	118.3 ± 13.7	122.1 ± 16.0
FVC, L	3.41 ± 0.92	4.30 ± 0.90	4.16 ± 0.98
Percent of predictive normal value	97.5 ± 20.5	122.9 ± 14.9	126.4 ± 17.1
FVC IN, L	3.48 ± 0.92	4.43 ± 0.91	4.25 ± 0.99
Percent of predictive normal value	95.4 ± 19.3	121.9 ± 14.1	124.2 ± 16.0
FEV1/FVC, %	41.0 ± 11.5	78.3 ± 4.2	79.5 ± 5.2
DLCO, mmol/min/kPa	4.59 ± 1.83	8.03 ± 1.89	7.82 ± 2.22
Percent of predictive normal value	53.4 ± 19.1	92.3 ± 15.9	94.0 ± 13.5
KCO, mmol/min/kPa	0.96 ± 0.31	1.43 ± 0.24	1.50 ± 22
Percent of predictive normal value	67.7 ± 23.1	100.8 ± 15.2	102.1 ± 16.9
ITGV, L	4.61 ± 1.28	3.12 ± 0.72	3.06 ± 0.76
Percent of predictive normal value	144.1 ± 34.0	99.4 ± 17.0	100.7 ± 20.4
RV, L	3.51 ± 1.12	2.11 ± 0.44	1.95 ± 0.41
Percent of predictive normal value	159.2 ± 46.8	96.5 ± 17.1	91.6 ± 18.8

COPD: chronic obstructive pulmonary disease; BMI: body mass index; VKA: vitamin K antagonist; MRC: Medical Research Council; FEV1: forced expiratory volume in 1 s; FVC: forced vital capacity; FVC IN: inspiratory forced vital capacity; DLCO: diffusion capacity of the lung for carbon monoxide; KCO: Krogh factor; ITGV: intrathoracic gas volume; RV: residual volume.

**Table 2 jcm-08-01116-t002:** Patient characteristics of the Leuven COPD cohort. Continuous data are presented as mean ± standard deviation.

Leuven COPD Cohort	COPD Stage			
	GOLD I	GOLD II	GOLD III	GOLD IV
Subjects, *n*	69	74	75	72
Age, years	64.1 ± 7.1	67.0 ± 71	67.6 ± 8.6	64.1 ± 8.1
Male, *n (%)*	52 (75.4)	55 (74.3)	59 (78.7)	56 (77.8)
Smoking pack years	44.3 ± 22.7	51.0 ± 24.4	50.2 ± 25.7	51.5 ± 28.3
BMI, kg/m^2^	25.9 ± 3.5	26.1 ± 4.8	25.2 ± 5.5	23.4 ± 5.6
VKA use, *n (%)*	5 (7.2)	10 (13.5)	7 (9.3)	5 (6.9)
MRC, points	1.6 ± 0.8	2.5 ± 1.0	3.1 ± 1.1	3.4 ± 1.2
Aortic calcification score, *points*	0.79 ± 0.87	1.20 ± 0.88	1.45 ± 0.98	1.50 ± 1.05
Lung function values				
FEV1, L	2.68 ± 0.57	1.72 ± 0.41	1.13 ± 0.22	0.69 ± 0.14
Percent of predictive normal value	90.4 ± 8.6	61.4 ± 7.5	39.9 ± 5.2	24.3 ± 4.1
FVC, L	4.26 ± 0.91	3.35 ± 0.77	2.91 ± 0.65	2.31 ± 0.71
Percent of predictive normal value	114.3 ± 13.3	94.1 ± 16.2	81.5 ± 14.0	64.4 ± 15.0
FEV1/FVC, %	63.2 ± 5.1	52.0 ± 8.1	39.6 ± 7.4	31.4 ± 6.5
VC, L	4.24 ± 0.90	3.38 ± 0.80	2.99 ± 0.65	2.41 ± 0.70
Percent of predictive normal value	109.5 ± 13.1	91.7 ± 15.7	81.0 ± 13.8	64.4 ± 14.6
DLCO, mmol/min/kPa	6.63 ± 2.22	4.89 ± 1.57	3.89 ± 1.40	3.04 ± 0.88
Percent of predictive normal value	74.6 ± 20.5	57.4 ± 15.7	46.1 ± 15.6	35.8 ± 9.8
KCO, mmol/min/kPa	1.12 ± 0.30	1.04 ± 0.30	0.91 ± 0.30	0.88 ± 0.28
Percent of predictive normal value	81.3 ± 21.2	76.0 ± 22.1	67.1 ± 21.4	63.9 ± 20.0
ITGV, L	4.09 ± 0.81	4.41 ± 1.07	5.22 ± 1.21	6.10 ± 1.45
Percent of predictive normal value	121.5 ± 19.3	132.4 ± 28.8	156.2 ± 33.5	183.1 ± 36.9
RV, L	2.72 ± 0.67	3.17 ± 0.80	3.89 ± 1.00	4.95 ± 1.29
Percent of predictive normal value	115.6 ± 23.1	134.7 ± 31.7	166.0 ± 41.7	212.7 ± 50.3
TLC, L	6.95 ± 1.26	6.55 ± 1.31	6.88 ± 1.32	7.36 ± 1.56
Percent of predictive normal value	108.0 ± 11.0	104.8 ± 16.6	110.4 ± 18.1	118.3 ± 20.2
Airway resistance, kPa/L/sec	0.34 ± 0.12	0.47 ± 0.14	0.57 ± 0.14	0.77 ± 0.20
Percent of predictive normal value	152.4 ± 54.8	214.5 ± 64.2	260.8 ± 63.7	349.4 ± 91.7
Specific airway conductance, 1/kPa*sec	0.78 ± 0.24	0.51 ± 0.16	0.35 ± 0.11	0.23 ± 0.07
Percent of predictive normal value	91.2 ± 28.4	60.0 ± 19.1	41.1 ± 13.1	26.7 ± 8.2
Mortality				
Died during follow-up, *n (%)*	16 (23.2)	27 (36.5)	44 (58.7)	47 (65.3)
Mortality unknown, *n (%)*	4 (5.8)	0 (0)	2 (2.7)	1 (1.4)
Follow-up, years	5.9 ± 2.0	6.1 ± 2.1	4.9 ± 2.7	4.3 ± 2.8

VC: vital capacity; TLC: total lung capacity.

**Table 3 jcm-08-01116-t003:** Effects of variables on dp-ucMGP and pDES in the ICE-Age cohort and the Leuven COPD cohort.

	**dp-ucMGP**			**pDES**		
**ICE-Age Cohort**	**Ratio**	**95% CI**	***p*-Value**	**Ratio**	**95% CI**	***p*-Value**
VKA use, yes	198.6%	138.7 to 274.0%	*p* < 0.0005	4.2%	−8.1 to 18.1%	*p* = 0.522
Presence of COPD, yes	36.8%	21.5 to 53.7%	*p* < 0.0005	15.6%	8.3 to 23.4%	*p* < 0.0005
Gender, female	−8.9%	−17.4 to 0.5%	*p* = 0.064	5.3%	−0.3 to 11.2%	*p* = 0.064
Age, per year	2.3%	1.6 to 3.0%	*p* < 0.0005	2.1%	1.7 to 2.5%	*p* < 0.0005
BMI, per kg/m^2^	3.6%	2.5 to 4.7%	*p* < 0.0005	−0.1%	−0.7 to 0.5%	*p* = 0.794
Smoking packyears, per year	0.1%	−0.1 to 0.3%	*p* = 0.489	0.1%	0.0 to 0.2%	*p* = 0.132
Dp−ucMGP, per 10%	-	-	-	1.0%	0.4 to 1.5%	*p* = 0.001
Lung function parameters						
FEV1, per 10 pp	−3.0%	−6.8 to 1.0%	*p* = 0.173	1.0%	−1.0 to 4.1%	*p* = 0.323
FVC, per 10 pp	−1.0%	−3.9 to 2.0%	*p* = 0.533	1.0%	−2.0 to 3.0%	*p* = 0.585
FVC IN, per 10 pp	−1.0%	−3.9 to 2.0%	*p* = 0.547	0.0%	−2.0 to 3.0%	*p* = 0.667
FEV1/FVC, per 10 pp	−1.0%	−1.0 to 6.6%	*p* = 0.199	1.0%	−3.0 to 5.0%	*p* = 0.607
DLCO, per 10 pp	−4.9%	−8.6 to −2.0%	*p* = 0.005	−3.9%	−5.8 to −1.0%	*p* = 0.004
KCO, per 10 pp	−3.0%	−5.8 to 2.0%	*p* = 0.076	−3.0%	−4.9 to −1.0%	*p* = 0.004
RV, per 10 pp	0.1%	−1.0 to 2.0%	*p* = 0.430	0.0%	−1.0 to 1.0%	*p* = 0.428
ITGV, per 10 pp	0.0%	−2.0 to 2.0%	*p* = 0.850	0.0%	−2.0 to 1.0%	*p* = 0.723
	**dp-ucMGP**			**pDES**		
**Leuven COPD Cohort**	**Ratio**	**95% CI**	***p*-Value**	**Ratio**	**95% CI**	***p*-Value**
VKA use, yes	141.1%	62.9 to 256.4%	*p* < 0.0005	5.4%	−9.3 to 22.5%	*p* = 0.485
Gender, female	14.3%	−13.1 to 50.4%	*p* = 0.336	5.0%	−5.4 to 16.6%	*p* = 0.360
Age, per year	1.3%	−0.2 to 2.8%	*p* = 0.079	3.0%	2.5 to 3.7%	*p* < 0.0005
BMI, per kg/m^2^	1.9%	−0.4 to 4.3%	*p* = 0.109	−0.5%	−1.3 to 0.4%	*p* = 0.294
Smoking packyears, per year	0.0%	−0.4 to 0.5%	*p* = 0.845	0.2%	0.0 to 0.3%	*p* = 0.065
Dp-ucMGP, per 10%	-	-	-	0.6%	0.2 to 1.1%	*p* = 0.004
Lung function parameters						
FEV1, per 10 pp	2.0%	−3.0 to 6.2%	*p* = 0.472	2.0%,	0.0 to 4.1%,	*p* = 0.027
FVC, per 10 pp	0.0%	−4.9 to 5.1%	*p* = 0.925	1.0%	−1.0 to 3.0%	*p* = 0.296
FEV1/FVC, per 10 pp	5.1%	−3.9 to 13.9%	*p* = 0.272	5.1%,	2.0 to 9.4%	*p* = 0.001
VC, per 10 pp	0.0%	−4.9 to 5.1%	*p* = 0.989	1.0%	−1.0 to 3.0%	*p* = 0.441
DLCO, per 10 pp	−4.9%	−10.4 to 1.0%	*p* = 0.075	−2.0%	−3.9 to 0.0%	*p* = 0.084
KCO, per 10 pp	−4.9%	−10.4 to 1.0%	*p* = 0.075	−2.0%	−3.9 to −0.02%	*p* = 0.048
ITGV, per 10 pp	−3.9%	−6.8 to −1.0%	*p* = 0.023	−2.0%	−3.0 to −1.0%	*p* = 0.002
RV, per 10 pp	−3.0%	−4.9 to −1.0%	*p* = 0.012	−1.0%	−2.0 to −1.0%	*p* = 0.001
TLC, per 10 pp	−9.5%	−15.6 to −3.0 %	*p* = 0.004	−4.9%	−6.8 to −2.0%	*p* = 0.001
Airway resistance, per 10 pp	0.0%	−2.0 to 1.0%	*p* = 0.472	0.0%	−1.0 to 0.1%	*p* = 0.114
Specific airway conductance, per 10 pp	3.0%	−1.0 to 7.3%	*p* = 0.139	2.0%	0.0 to 3.0%	*p* = 0.009

COPD: chronic obstructive pulmonary disease; BMI: body mass index; VKA: vitamin K antagonist; FEV1: forced expiratory volume in 1 s; FVC: forced vital capacity; FVC IN: inspiratory forced vital capacity; VC: vital capacity; DLCO: diffusion capacity of the lung for carbon monoxide; KCO: Krogh factor; ITGV: intrathoracic gas volume; RV: residual volume; TLC: total lung capacity; 95% CI: 95% confidence interval; pp: percent points. Table 3 represents expected change (‘Ratio’) of dp-ucMGP and pDES with 95% CI, when the variable in the first column increases by 10%, by 10 percent points or by 1 unit.

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
