# Peer review of "Low Vitamin K Status Is Associated with Increased Elastin Degradation in Chronic Obstructive Pulmonary Disease"

_jcm, 2019, doi:10.3390/jcm8081116_

Reviewer 1 Report

The mauscript is well written, the concept of the study is correct, the flaws are honestly described in limitness of the study section. Well done and congratulations!

Minor remark: the level of vitamin K should be measured directly in serum n.e. by HPLC in el least 10% of study participants, the results sholud be calculated and compared with Matrix Gla Protein. If the results of both methods would be positevely correlated, it would be an arrgument and explanation for choise  of cheaper of better available but not direct method for vitamin K status evaluation.

Author Response

(x) Extensive editing of English language and style required   

Response 1 - The revised manuscript has been checked by a professional English editing service.  

Point 2 -   The manuscript is well written, the concept of the study is correct, the flaws are honestly described in limitness of the study section. Well done and congratulations!

Response 2 - We would like to thank the referee for the compliments.

Point 3 -   Minor remark: the level of vitamin K should be measured directly in serum n.e. by HPLC in el least 10% of study participants, the results should be calculated and compared with Matrix Gla Protein. If the results of both methods would be positively correlated, it would be an argument and explanation for choice of cheaper of better available but not direct method for vitamin K status evaluation.

Response 3 - We would like to thank the reviewer for the suggestion. However, circulating vitamin K levels reflect recent dietary vitamin K intake but not vitamin K status in tissues. The correlation between circulating vitamin K and dp-ucMGP levels is poor. Another issue would be the question which form of vitamin K in serum should be measured: phylloquinone, dihydrophylloquinone, or any of the menaquinones (MK-4, MK-5, MK-6, MK-7, MK-8, MK-9, MK-10, MK-11 or MK-12)? It is not known how and to which extent these various K-vitamers contribute to tissue vitamin K status. The serum concentration of most of these menaquinones is insufficient to be detected, whereas they can be high in tissues. Also, the tissue distribution of these menaquinones is insufficiently known. Hence measurement of circulating forms of vitamin K is not very useful, which has been stated by many experts in the vitamin K-field, such as Cees Vermeer.

Reviewer 2 Report

Manuscript: JCM-543802 Low vitamin K status associates with increased elastin degradation in Chronic Obstructive Pulmonary Disease

This study described the association between vitamin K status, assessed by inactive MGP levels, and elastin degradation, measured by plasma desmosine levels, in patients with COPD.  The study found that vitamin K status was inversely related with elastin degradation, suggesting that reduced vitamin K levels in COPD may contribute to the disease pathogenesis.  Interestingly, few associations between vitamin K status and lung function parameters were found, although mortality was higher in patients with the lowest vitamin K status.  Overall, the basis for the present study was sound and filled an as-of-yet unanswered question.  The methods appear to be sound and the conclusions are supported by the data reported.

Major Comments:

1)      A surprising finding was that vitamin K status was not associated with most lung function parameters, which would have been expected.  Despite the paucity of data in the current literature, perhaps the authors might provide some thoughts as to why this association was not observed?  Of course, this would be speculative, but it might be helpful to the reader to understand some possible explanations, even if they are hypothetical.

2)      Were patients stratified by COPD phenotype, that is, emphysema vs. chronic bronchitis phenotype?  It could be expected that elastin degradation would be more prominent in an emphysema phenotype compared to a similar “severity” patient with more of a chronic bronchitis phenotype.

3)      Is it possible to assess MGP activity?  This would aid in linking the proposed mechanism, given that low vitamin K levels do not necessarily mean lower MGP activity in vivo.

Minor Comments:

·         It could be beneficial to consider move the comments on the role of elastin in COPD from Lines 274-278 in the discussion to the introduction to help “close the circle” describing the relationship between vitamin K, elastin, and COPD pathogenesis.

·         It is unclear whether vitamin K status and elastin degradation were assessed only once in the Leuven cohort, despite the availability of longitudinal follow-up data.  Is this the case?  That is, was there only a single cross-sectional assessment of vitamin K status and elastin degradation, but longitudinal lung function in this cohort?  Otherwise, if longitudinal vitamin K and elastin degradation are available to correlate with lung function, that data would be extremely interesting and novel

·         Were dietary records or questionnaires collected?  These might have been useful in disentangling cofounding effects of diet on vitamin K status in this population

·         Was HRCT data available for either cohort?

·         Would it be possible to include a table with multiple linear regression data (e.g., coefficients, beta-values, and p-values)?

·         Table 1a typo – italicize “n” following “Former smokers” for consistency

·         Table 1a FVC percent predictive normal value and subsequent line (FEV1/FVC ratio) – check formatting for third column (never-smoking controls) – appears to be misaligned compared with other columns

Author Response

Point 1 - English language and style
(x) English language and style are fine/minor spell check required   

Response 1 - The revised manuscript has been checked by a professional English editing service.  

Point 2 - This study described the association between vitamin K status, assessed by inactive MGP levels, and elastin degradation, measured by plasma desmosine levels, in patients with COPD.  The study found that vitamin K status was inversely related with elastin degradation, suggesting that reduced vitamin K levels in COPD may contribute to the disease pathogenesis.  Interestingly, few associations between vitamin K status and lung function parameters were found, although mortality was higher in patients with the lowest vitamin K status.  Overall, the basis for the present study was sound and filled an as-of-yet unanswered question.  The methods appear to be sound and the conclusions are supported by the data reported

Response 2 - We would like to thank the referee for the compliments.

Major comments

Point 3 -   A surprising finding was that vitamin K status was not associated with most lung function parameters, which would have been expected.  Despite the paucity of data in the current literature, perhaps the authors might provide some thoughts as to why this association was not observed?  Of course, this would be speculative, but it might be helpful to the reader to understand some possible explanations, even if they are hypothetical.

Response 3 - We demonstrated that vitamin K status was associated with accelerated elastin degradation. Elastin degradation is a driver of disease progression in COPD (i.e. lung function decline) rather than a marker of disease severity (such as lung function tests and COPD GOLD stage). We therefore expect that vitamin K status is related to disease activity instead of severity. We would expect to find accelerated lung function decline in subjects with low vitamin K status. Unfortunately, follow-up lung function data were not obtained in both cohorts, so it was not possible to test whether this assumption is correct. In the discussion section (lines 334-340), we clarify this theory which might be an explanation for the scarcity of associations between lung function parameters and vitamin K status.Furthermore, there might have been an association between vitamin K status and parenchymal destruction, which we could not assess since CT-scans were lacking.

Point 4 - Were patients stratified by COPD phenotype, that is, emphysema vs. chronic bronchitis phenotype?  It could be expected that elastin degradation would be more prominent in an emphysema phenotype compared to a similar “severity” patient with more of a chronic bronchitis phenotype.

Response 4 - We agree with the reviewer that this would be an interesting question. However, CT scans were not available for patients in the COPD cohorts. Unfortunately, it was not possible to stratify patients by COPD phenotype. Additional studies are therefore needed. 

Point 5 - Is it possible to assess MGP activity?  This would aid in linking the proposed mechanism, given that low vitamin K levels do not necessarily mean lower MGP activity in vivo.

Response 5 - We agree with the reviewer that it would be interesting to assess MGP activity. However, unfortunately, it is not possible to assess MGP in a functional assay (Cees Vermeer; vitamin K expert). 

Minor Comments:

Point 6 - It could be beneficial to consider move the comments on the role of elastin in COPD from Lines 274-278 in the discussion to the introduction to help “close the circle” describing the relationship between vitamin K, elastin, and COPD pathogenesis.

Response 6 - We agree that these sentences are better placed in the introduction. Therefore, the concerning section has been moved from the discussion to the introduction. 

Point 7 - It is unclear whether vitamin K status and elastin degradation were assessed only once in the Leuven cohort, despite the availability of longitudinal follow-up data.  Is this the case?  That is, was there only a single cross-sectional assessment of vitamin K status and elastin degradation, but longitudinal lung function in this cohort?  Otherwise, if longitudinal vitamin K and elastin degradation are available to correlate with lung function, that data would be extremely interesting and novel

Response 7 – We fully agree with the referee that it would be interesting to correlate longitudinal vitamin K and elastin degradation data with lung function. However, vitamin K status, elastin degradation rate and lung function tests, were only assessed once. Additional studies are needed to assess these associations.  

Point 8 - Were dietary records or questionnaires collected?  These might have been useful in disentangling cofounding effects of diet on vitamin K status in this population

Response 8 - We agree with the reviewer that dietary records could help to disentangle confounding effects of diet on vitamin K status. However, no dietary records or questionnaires were collected in the cohorts that we used.  

Point 9 -  Was HRCT data available for either cohort?

Response 9 – Unfortunately, no HRCT data were available for both cohorts. 

Point 10 - Would it be possible to include a table with multiple linear regression data (e.g., coefficients, beta-values, and p-values)?

Response 10 - We would like to thank the referee for the suggestion. We added a table with multiple linear regression data (table 2).

Point 11 - Table 1a typo – italicize “n” following “Former smokers” for consistency

Response 11 - The concerning “n” has been italicized. 

Point 12 - Table 1a FVC percent predictive normal value and subsequent line (FEV1/FVC ratio) – check formatting for third column (never-smoking controls) – appears to be misaligned compared with other columns

Response 12 - The alignment has been corrected.

Reviewer 3 Report

Piscaer submitted an interesting mansucript concerning COPD and Vit K status.

They found that vitamin K status was inversely associated with progression of COPD.

The finding of the mansucript was new and many readers will be interested in.

Author Response

Point 1 - English language and style:
(x) English language and style are fine/minor spell check required   

Response 1 - The revised manuscript has been checked by a professional English editing service.  

Comment 2 - Piscaer submitted an interesting manuscript concerning COPD and Vit K status.

They found that vitamin K status was inversely associated with progression of COPD.

The finding of the manuscript was new and many readers will be interested in.

Response 2 - We would like to thank the referee for the compliments.